# Science Skills Development through Problem-Based Learning in Secondary Education

Jorge Pozuelo-Muñoz [1] , Elena Calvo-Zueco [2] , Ester Sánchez-Sánchez [2] and Esther Cascarosa-Salillas [1,*]

1   Department of Specific Didactics, University of Zaragoza, 50009 Zaragoza, Spain; jpozuelo@unizar.es
2   High School Internacional Ánfora, 50410 Zaragoza, Spain; ecalvo@colegiointernacionalanfora.com (E.C.-Z.); esanchez@colegiointernacionalanfora.com (E.S.-S.)
*   Correspondence: ecascano@unizar.es

**Abstract:** We present a study carried out with 16-year-old students in Spain using a problem-based learning approach as a pedagogical mode to develop science skills. The main objective of this work was to analyze the development of science skills through an inquiry process in class. The data were collected through audio and video recordings. The students were given the freedom to choose a problem to solve, and they decided on a near-environmental problem to research. They suggested a research question, formulated a hypothesis, designed experiments, observed, collected data, and searched for information. The teacher acted as a facilitator of resources. Finally, the students communicated the results obtained in their inquiry process. They performed all the above while asking themselves questions they had to answer during the course of the project, which increased in depth as the work evolved. The results of this research present PBL as an optimal methodology to develop scientific skills, such as inquiry practice, by means of asking questions.

**Keywords:** PBL; inquiry; science skills; science motivation; secondary education

## 1. Introduction

Over the years, some researchers have explored the low number of scientific vocations among students, which results in poor scientific literacy in society [1]. This, in turn, leads to a lack of critical thinking (necessary in daily life) and a strong vulnerability to fake news, among other things [2]. Basic scientific education is necessary for many situations in daily life, and, for that reason, there is a need for real scientific literacy that is increasingly rich in scientific and technological content and educates the public in a social context.

Two of the most important factors determining scientific interest are self-confidence and motivation [3], which also influence efficiency in school science. The authors in [4,5] presented results of the low rates of self-confidence in science learning in young Europeans, which leads to low interest in the subject. Moreover, ref. [6] demonstrated that the methodologies used in the teaching of sciences influenced attitudes toward them, thereby establishing the connection between students' attitudes and their later scientific interests. In conclusion, all of the above should be taken into account when designing the teaching of sciences in school, and, according to Solbes et al. [7], this is not yet widely done. In view of all this, it seems clear that science classes should be designed with the aim of helping students investigate their scientific concerns while learning what teachers have to teach. Prior studies reveal that students are interested in solving problems in their immediate environment, that is, if the scientific problems are contextualized in the students' environment, their interest grows [8–11]. In this respect, knowing whether students develop science process skills should be a topic that teachers assess. Therefore, an ideal teaching methodology could consist of students themselves posing problems contextualized in their own environments and based on their own interests [12].

Secondary school teachers should teach content, but also procedures [13]. In this regard, authors like Bevins and Price [14], Jiménez-Aleixandre and Crujeiras [15] and

Mosquera Bargiela et al. [16] believe that scientific practices (inquiry, argumentation and modelling) are methods for teaching science using problem-based learning (PBL) [17], which helps to promote research skills in students and the internalization of new knowledge. For example, González Rodríguez and Crujeiras Pérez [18], Navy et al. [19] and Osborne and Dillon [20] believe in the usefulness of working through PBL to develop the inquiry practice used in experimental activities.

PBL is an educational instruction method, created by Dewey [21]. Content knowledge and problem-solving skills are the goals of this learning vehicle [22]. According to authors such as Hmelo-Silver [23] and Merrit et al. [24], the goals of PBL are grouped into: (a) content knowledge (construct an extensive and flexible knowledge base, academic achievement, knowledge retention, conceptual development), (b) procedural knowledge (develop effective problem-solving skills and self-directed, lifelong learning skills), (c) become effective collaborators, and (d) attitudes (become intrinsically motivated to learn, engaged). PBL is focused on learning through problem-solving and by the integration and application of knowledge in a real-world setting [25], allowing, as a consequence, the development of competencies and skills [26]. Drake and Long [27] investigated the usual PBL design, addressing eight components: problem, small group, students-centered iterative inquiry process, resources, technology, partnership with community, communication of findings, and teachers' roles as facilitators. There are not many published studies on the benefits of PBL in teaching science to secondary school students. Some of these studies offer promising results, such as that PBL favors the development of students' critical thinking [28] and also helps teachers and students learn the practices of scientists [29].

It must be taken into account that the problem is the focus of the learning process, acting as the stimulus for students' motivation and activity [30]. Thus, the problem should be complex, relating to real life and ill-structured, in order to offer students free inquiry and open-ended solutions in a wide range [24,31]. On the other hand, several authors indicated that students should be given the autonomy to discover their own problems and solve them [30,32]. Once the problem is defined, students work collaboratively in small groups [22,29,31], centered in an iterative inquiry process that is greatly promoted by the PBL instructional model [30].

Inquiry is an intentional process of diagnosing problems that requires identifying assumptions, applying logical and critical thinking, and considering alternative explanations [33], and it is also directly related to how scientists study the natural world and propose explanations based on the evidence stemming from their work [34]. According to Pedaste et al. [35] the phases of inquiry are: orientation, experiment design, investigation, conclusion, communication of results and discussion. Inquiry is based on questions that are asked (or self-asked) at the beginning and act as generators and organizers of knowledge [36]. This questioning arouses the desire to find out new things and helps individuals to reflect on their own knowledge and the learning process [15]. In 1994, Graesser and Person [37] classified the type of questions that students could ask into shallow, intermediate and deep. They related the type of question asked by students to the level of reasoning required to ask said question. Consequently, questions with the lowest quality (shallow) are related to verification, comparison, and completing a concept or definitions; reasoning, with an intermediate quality, involves giving examples, interpreting, specifying concrete aspects and quantifying; and finally, deep reasoning appears when questions establish a causal antecedent or consequence, or the orientation towards a goal or expectation. It is possible to conclude the problem-based learning process by analyzing the type of question that the students formulate throughout the inquiry process [28,38].

According to Harlen [39], the most interesting questions students may ask in the learning process are searchable issues, namely, questions that can be answered through research.

Inquiry through experimentation is part of the process for preparing models in the school context in the phases of preparing and testing mental models. It is aimed at solving practical challenges, which is very useful for the procedural understanding of science—that is, understanding the processes that characterize research [18]. In the same vein, according

to Hodson [40], students like to know what they are doing because not knowing unsettles them, and they appreciate cognitive challenges as they are able to answer questions for themselves. This means that the tasks that are designed should be suitable for helping students to have enough control and independence, without this interfering in the learning process.

Even today, the development of the PBL methodology based on the student's interest, as a method of teaching science through the practice of inquiry, is not common in Spanish secondary education classrooms. On the other hand, there are real difficulties in teacher training for evaluating student learning through this type of methodology.

Taking the above into account, this work proposes research to evaluate the development of students' science process skills in inquiry by asking questions in a PBL context. That is, we evaluate the second objective of the PBL through the analysis of the type of questions that the students ask themselves in the inquiry process.

Our research questions are: Does the PBL methodology facilitate the development of science skills in class? What type of questions do students ask themselves in a science inquiry practice through problem-based learning designed based on their own interests?

## 2. Materials and Methods

### 2.1. PBL Procedure

For the design of our PBL process, we attended to the components described by Merrit et al. [24], which resulted in two interactions (phase 1 and phase 2), as described later in the results section.

The problem: Over one academic year, the teachers used a problem-based model of learning, using question-based inquiry as a vehicle tool. This year-long project was developed within the context of the scientific culture subject, which focuses on educating students to help them to understand the environment in which they live, by providing them with tools to obtain answers to everyday questions. Considering the concerns that the students showed regarding issues in their environment, specifically, they were interested in discovering the process through which the manure from the farms close to the school could be used as fertilizer for plants, given that the accumulation of this waste is a real and imminent environmental problem. Therefore, they suggested researching the use of said waste as fertilizer for vegetables, attempting to answer the following questions: Can manure be used as a fertilizer? What is the best proportion for plant growth? Identifying problems from their real context is considered important [41] as it acts as a motivation source because students feel their work is useful in their nearby community. Starting from this research question, the students had to design the research to find out whether manure can be used and is effective as a fertilizer for plants. Subsequently, they had to conclude if this use could be a viable solution to the environmental problem of manure waste.

The small group: The work was conducted with 10 students (50% girls) aged between 16 and 17 years old, working in a collaborative way. Working in collaborative groups allows students to be engaged in building knowledge, which is shared among them [30].

The iterative inquiry process: The search for solutions to the problem implies the development of an inquiry process based on the formulation of questions. These self-formulated questions, not provided by the teacher, activate their desire for knowledge, facilitate the understanding of new concepts, help them build sequenced knowledge, and arouse their epistemic curiosity. The results section describes this student inquiry process in detail.

Resources and technology: It is important to allow students to make their own decisions during their investigation, including what information they need to locate or how to analyze and evaluate the information to solve the problem [30]. Therefore, the students had access to the laboratory to be able to carry out the experiments they designed to prove or refute their hypotheses. The computers were freely accessible throughout this stage for the free consultation of information.

Partnership with the community: Since the problem that the students raised was related to their immediate environment, the learning process was expanded so they could have direct contact with farmers in the area, as well as other members of the community, such as experts in the chemical industry (fertilizer industry).

Communication of findings: Once the process was finished, the students communicated the results to their classmates, their teachers, and the university teachers who acted as researchers in this work.

Teachers as facilitators: The teaching staff acted as teacher-researchers, collecting the questions that students asked in each phase of the project and, as support, providing access to what was required. As Mosquera Bargiela et al. [16] demonstrated, showing an attitude of support and providing resources to students so that they can answer in the school context encourages their scientific development.

## 2.2. Assessment/Evaluation

The knowledge the students acquired was assessed using several tools, as recommended by Brenneman [42] and García-Carmona et al. [43]. The project sessions were audio- and video-recorded, as collecting audio and video data helps teachers to analyze the comments pupils make when working in a group, the answers to their questions, and their reactions to several situations; in other words, this tool should be used as much as possible [44]. The teachers kept an observation report in which they recorded the questions that the students asked in each phase of the project. Photos were taken of both the procedures followed and the results of the fertilization tests.

## 3. Results

The students started with a problem to solve (choose by themselves), and they had to design the procedure in order to obtain results and draw conclusions about this problem. Therefore, one of the most relevant results is the design that the students prepared in order to reach a conclusion about the question—in other words, the development of the PBL phases. As the students had decided the problem, this work relates to the fourth stage presented by Arici and Yilmaz [45], that is, looking for a solution for the problem.

## 3.1. Phase 1

In the first phase of the project, the students wanted to find out if manure was viable as a fertilizer and at what dose. To discover this, they suggested an experimental study where the variable was the proportion of the fertilizer in an aqueous solution. They subsequently developed their own analytical method based on the existing study "Impact of Artemisia absinthium hydrolate extracts with nematicidal activity on non-target soil organisms of different trophic levels" [46], which consists of measuring the elongation of the roots of an onion bulb (Allium) in a test tube containing water and nutrients. When the onion bulb was rehydrated, there was a stimulation in the growth of cells, which, in turn, enabled the growth of the roots.

The students contacted cattle farmers in the area, who provided them with solid manure (sun-dried pig manure). The first problem they had to solve was the change in the form of the manure, as the test, as described in the procedure, was performed in a liquid medium. After finding relevant information, they dissolved the manure by washing the solid with water. Then, they filtered the water and repeated the washing process five times to achieve greater concentration. This filtered liquid was considered to be at 100% fertilizer concentration, and solutions in water were prepared from that 100% sample.

They prepared five different concentrations of manure in water: 100%, 75%, 50%, 25%, and 0%. The students decided that, given that there could be statistical dispersion in the results, they would perform eight repetitions for each test and would consider the mean value of all repetitions as valid.

After filling all the test tubes with the appropriate concentrations, they prepared the onion bulbs: they peeled 14/21-size onions and put them in the tube with the solution.

When they finished preparing the test, they placed the tubes in an incubator (heater) for 2 weeks at 25 °C to replicate the ideal atmosphere for the growth of this type of onion.

After this period had passed, the students measured the length of the onion roots (Table 1).

**Table 1.** Growth (in cm) of the onion roots after the experimentation in phase 1.

| % Fertilizer | 0 | 25 | 50 | 75 | 100 |
|---|---|---|---|---|---|
| Tube 1 | 2.6 | 2.8 | Fail | 1.9 | 1.4 |
| Tube 2 | 0.6 | 1 | 1.4 | 0.5 | 1.2 |
| Tube 3 | 3.2 | 0.7 | 0.5 | 1.3 | 1 |
| Tube 4 | 1.5 | Fail | Fail | Fail | 0.6 |
| Tube 5 | 3.5 | 3.6 | Fail | 1.8 | 0.5 |
| Tube 6 | Fail | Fail | 2.3 | 0.2 | 0.7 |
| Tube 7 | Fail | Fail | 0.6 | 2.6 | 0.8 |
| Tube 8 | Fail | Fail | 1 | Fail | Fail |
| Mean value | 2.3 | 2.0 | 1.2 | 1.4 | 0.9 |

The students analyzed the results obtained: "After performing the experiment, we observe that some onions have not grown. We call these experiments a fail. We attempt to discover why these samples did not grow. To do this, we review the process of introducing the samples in the tubes and realize that some of them were very small and did not make contact with the manure solution. We believe that the roots did not grow because of poor contact with a solution. At the same time, we also observe that some onions were poorly peeled, and this could be another determining factor in their failure to grow.

We will now analyze the tests that did not fail and managed to grow. As we performed 8 tests with each percentage of manure, the final result (mean value) is more accurate. We can see that the highest mean growth corresponds to 0% manure, while the lowest is the one fertilized with 100% manure. We also observe that the highest growth has been with 0% of manure, which leads us to conclude that manure used without any other type of fertilizer has a very low productive effect.

Before approving the results, the students questioned the data by reflecting on what they had obtained and attempting to find an explanation: "When we analyzed the results we were surprised that the best data were obtained with a 0% concentration of manure. After reflecting on these data, we concluded that using too much manure causes the crop not to grow. The cause could be that too much manure results in a large amount of nitrogen in the soil/water and this prevents the crop from absorbing the nutrients correctly, thus hindering its growth. However, we are aware that these results go against practice in the field, because we asked around and rain-fed crops in the local area were only fertilized with farm waste, that is, the manure used in our tests. Therefore, we put forward a new hypothesis, that the conclusions were wrong." The students considered how to check this new premise. After discussing the possible alternatives, they decided to follow two paths. Firstly, they consulted an agricultural fertilizer company. They arranged a meeting and explained the results of the tests to the technical manager, who told them that the first root of an onion is weak and contact with a fertilizer with a high proportion of nitrogen could have damaged the roots in the samples. The students concluded: "before the following batch of experiments, we will leave the bulbs at least one week in water, before putting them in contact with the fertilizer".

Secondly, and at the same time, the students sought an answer from professionals from "We are Scientists"—a program financed by the Spanish Foundation for Science and Technology (FECYT) that offers students the chance to talk to scientists from all fields. In this context, the students summarized the information obtained as follows: "After discussing the results with young scientists, they informed us that if the plant has access to nearby nutrients its roots do not develop very much. In contrast, in treatments without fertilizer, the plant explores in order to find nutrients and, therefore, develops longer roots.

That is why root length is not the factor that determines greater development of the plant, it is just an indicator of how hard it is for the plant to find nutrients. Consequently, we need to observe the effect on the root mass and on the aerial part of the plant".

Taking this information into account, the students reflected on the results obtained once again and concluded that this argumentation explained what they had observed, as the higher concentrations of nutrients produced the shortest roots (concentrations from 50% to 100%), while the roots were longer with lower concentrations (0% and 25%). However, they could not check the development of the bulbs because they had disposed of those samples. Therefore, they decided to create a new design for experiments and established that they would measure the growth of the plant by measuring the length and mass of the stalk, bulb and root. They also introduced a new fertilizer (an industrial rooting agent) to compare the results against those from the manure in the first design.

*3.2. Phase 2*

In this second phase of the project, taking into account the conclusions obtained in the first phase, the students decided to analyze the growth of the onion bulbs by comparing a commercial fertilizer (using the manufacturer's recommended concentration) and a 25% aqueous solution of the manure used in the first phase. They chose this proportion as it was the most similar to the dilution of the commercial fertilizer.

Before the experiments, based on what they had learned in phase 1, the students let the onion bulbs grow for two weeks using water as the only nutritional sustenance. In these experiments, they started with bulbs without stalks or roots. Then, they performed eight tests with each of the fertilizers and obtained two fails with each fertilizer. Therefore, there were six valid results with each fertilizer. These data are shown in Table 2 and in Figure 1.

**Table 2.** Results after the experimentation in phase 2, with fertilizer A (25% manure solution) and fertilizer B (commercial rooting agent).

| | Fertilizer Type | Stalk Mass (g) | Bulb Mass (g) | Root Mass (g) | % Stalk Mass | % Bulb Mass | % Root Mass | Stalk Length | Root Length |
|---|---|---|---|---|---|---|---|---|---|
| Tube 1 | A | 1.480 | 1.027 | 0.603 | 47.6 | 33.02 | 19.4 | 25 | 9 |
| | B | 2.001 | 0.400 | 2.00 | 45.47 | 9.09 | 45.44 | 30 | 11 |
| Tube 2 | A | 1.610 | 1.013 | 0.640 | 49.34 | 31.05 | 19.61 | 30 | 10 |
| | B | 1.330 | 1.330 | 0.600 | 40.80 | 40.80 | 18.40 | 31 | 9 |
| Tube 3 | A | 2.400 | 2.070 | 1.461 | 40.47 | 34.90 | 24.63 | 25 | 15.5 |
| | B | 0.810 | 1.970 | 0.880 | 22.13 | 53.83 | 24.04 | 15 | 10 |
| Tube 4 | A | 1.900 | 1.600 | 0.560 | 46.80 | 39.41 | 13.79 | 18 | 9 |
| | B | 1.250 | 2.130 | 1.580 | 25.20 | 42.94 | 31.86 | 7 | 9 |
| Tube 5 | A | 1.800 | 1.400 | 0.720 | 45.92 | 35.71 | 18.37 | 22 | 10.2 |
| | B | 1.120 | 1.400 | 1.670 | 26.73 | 33.41 | 39.86 | 17 | 10.7 |
| Tube 6 | A | 1.700 | 1.500 | 0.630 | 44.39 | 39.16 | 16.45 | 19 | 9.7 |
| | B | 1.860 | 1.700 | 1.100 | 39.91 | 36.48 | 23.61 | 27 | 9.5 |
| Mean Value | A | 1.815 | 1.435 | 0.769 | 45.75 | 35.54 | 18.71 | 23.2 | 10.6 |
| | B | 1.395 | 1.488 | 1.305 | 33.37 | 36.09 | 30.54 | 21.2 | 9.9 |

The students reflected on the results they had obtained: "When we compared the mean values of the tests performed with the two fertilizers, we did not observe significant differences, except in the value of the root mass. With the commercial rooting agent, the weight of the root as part of the whole plant represented a much higher percentage than with the manure. In other words, in proportion to the plant as a whole, the mass of the root

increased much more with the commercial fertilizer. We believe that the similar increase observed in root and stalk length and in bulb mass in both tests means that the plant has had sufficient nutrients for growth and did not have to use the bulb's reserves.

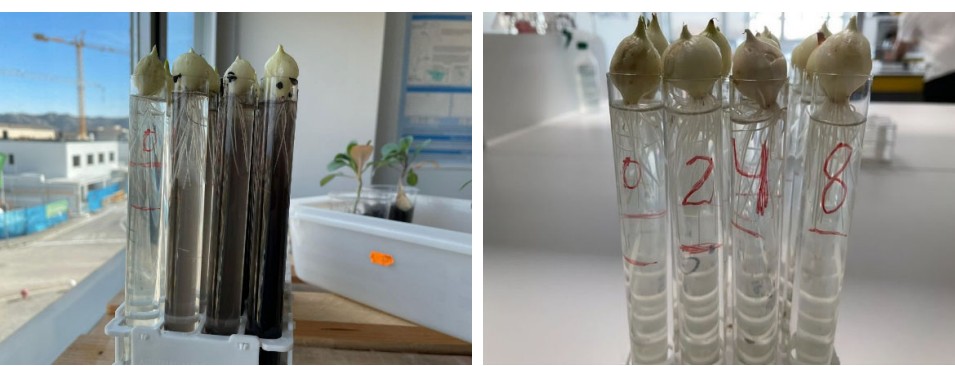

**Figure 1.** Growth of the root of A. cepa bulbs after phase 2.

We can also see that the root length is very similar with both fertilizers. As we have learnt, very long roots mean that the solution is not providing sufficient nutrients to the plant and, consequently, the roots grow longer to reach farther and attempt to get more nutrients. In contrast, when the solution contains sufficient nutrients, the roots are thicker and shorter. Taking this into account, if we observe the root mass, we can see that the mean value of the data obtained using the commercial fertilizer is higher; specifically, the bulbs fertilized with B have 76% more mass. We also observe that the increase in the stalk mass is greater in the bulbs that have been fertilized with manure, with approximately 30% more mass. This could mean that the nutritional content of the manure is higher than that of the rooting agent.

Figure 2 portrays the steps taken in the context of the PBL in each of the two phases.

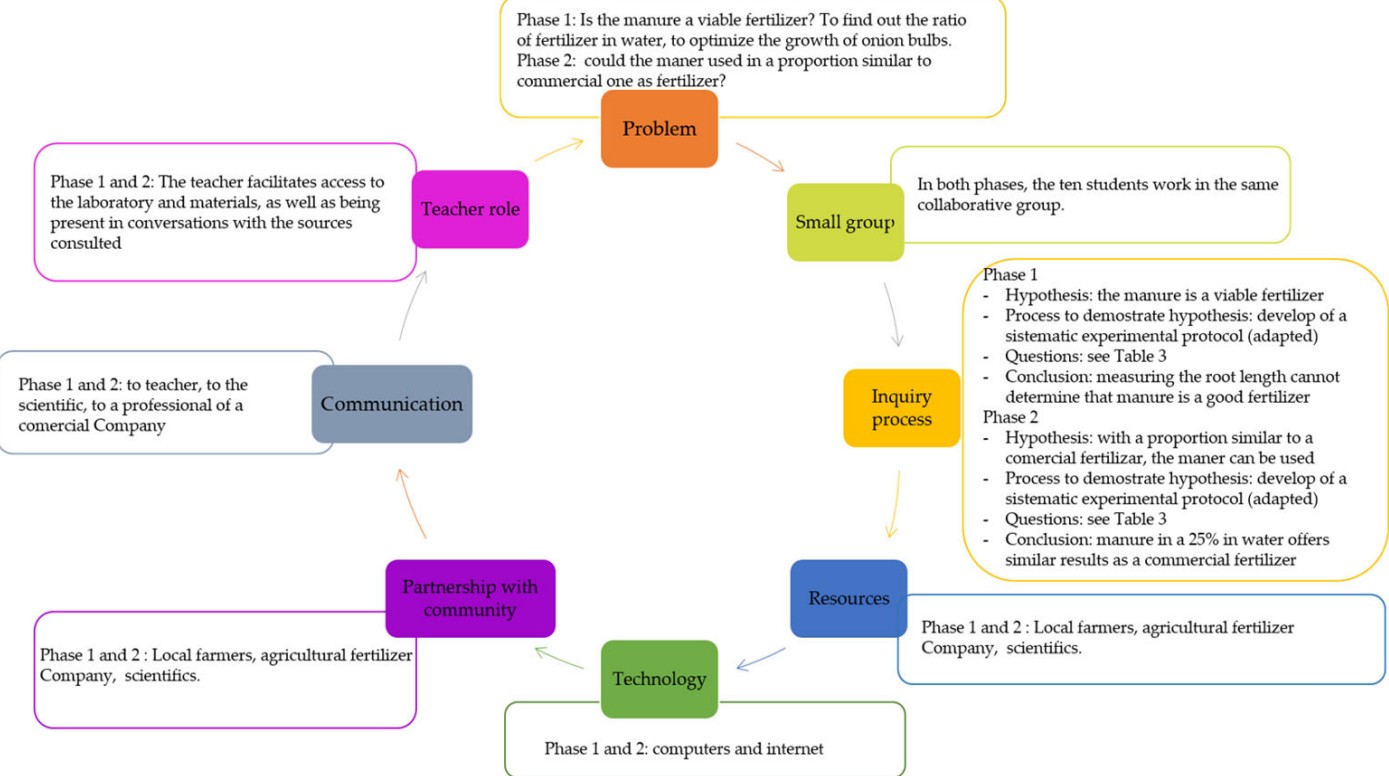

**Figure 2.** Steps of the PBL process.

Summarizing the science skills identified during the two PBL phases developed, the students identified a problem in their environment, established hypotheses, designed a laboratory protocol in which they took measurements, compared, experimented, identified researchable questions, sought guidance in different scientific sources, compiled results and diffused conclusions.

On the other hand, the questions that the students asked themselves during the process were analyzed and classified into categories according to [38]; these can be observed in Table 3. The questions formulated in a way that they can be answered with "yes" or "no" are shallow questions, usually about verification or comparison. When the questions, despite having to be answered with "yes" or "no", require a connection of ideas, or a detailed analysis, such as previous experimentation, they are considered intermediate questions. This includes questions about quantification and questions for which the students do not expect to find a direct answer. These questions typically begin with "is it possible?", "how many" or "what is". Finally, deep questions contain more than one aspect to be considered, and complex relationships between these aspects, including solution-oriented questions about environmental issues. Deep questions begin with "why" or "how".

**Table 3.** Categories of questions asked by students during the process.

| | | Before the Project | Phase 1 | Phase 2 | End of the Project |
|---|---|---|---|---|---|
| Shallow questions | Verification | - Can we research an environmental problem in our environment? <br> - Can manure be used as a fertilizer? | - Are our results correct? | | |
| Intermediate questions | Specific concrete aspects | | | - Is it possible that each fertilizer boosts the growth of one part of the plant? <br> - Will it be useful for us to compare the manure solution data with another commercial fertilizer? | |
| | Quantifying | | - What is the best proportion for plant growth? <br> - How many repetitions should we do in order for the result of each test to be representative? | | |
| Deep questions | Establish causal antecedent | | - Why have not all the bulbs grown? <br> - Why have the best results been obtained with the 0% manure solution? | | |
| | Establish consequence | | | | - Could all livestock waste be used as agricultural fertilizer? <br> - Would there be enough to replace commercial fertilizer? |
| | Orientation towards a goal or expectation | -How can we research into the environmental problem of manure? | - How can we apply solid manure to bulbs? <br> - How can we check our results? | - How can we measure which of the two is the best fertilizer? | |

After completing the project, the students spent one week presenting their results to the teaching staff of the school and to the university lecturers. In addition, in this final phase, they asked new questions for future research: "Would all types of animal waste be

suitable as fertilizers?", "In a long term, is an animal fertilizer or a commercial one better?", "Could pruning waste be used as a natural fertilizer?".

Despite not having been analyzed in detail, the results collected by the researchers can conclude that the students maintained a high level of motivation throughout the course. The students worked on issues related to the problem in classes for subjects other than the one in which they developed this problem; they met to work after school hours, discussed the issue with other students who were not part of the project; and asked the teacher to dedicate more hours to this work. The students expressed their desire to continue investigating the research question throughout the year, so this motivation maintained over time led to intrinsic interest.

## 4. Discussion and Conclusions

In the present work, we have developed a PBL methodology to work on inquiry in science classes in order to develop science process skills.

It is known that science teaching methodologies in secondary education do not favor the development of scientific skills; in the best of cases, the students follow guided practice in the laboratory without needing to formulate hypotheses or ask themselves researchable questions. This type of science teaching does not develop critical thinking, nor does it establish enduring scientific knowledge. If we want our students to develop these skills, we have to design contexts that help them identify a problem to solve, from which secondary questions arise that must be solved through the inquiry process to reach a final conclusion. Therefore, in this work, we start from an initial problem to be solved and analyze whether the PBL methodology, which contextualizes the work, encourages the students to ask themselves researchable questions that expand their knowledge about the initial problem posed. Throughout the process, the students raised the problem to be solved, formulated hypotheses, and designed a laboratory procedure to respond to their hypothesis. Throughout the development, several questions were formulated that broadened the knowledge necessary to solve their initial problem.

A second aim was to analyze the kind of questions students ask in a context of inquiry in order to assess if this methodology allows the students ask themselves relevant questions, from the point of view of the development of scientific skills.

Summarizing our results, the PBL methodology followed facilitates the learning of science, and in this specific case, the acquisition of scientific skills. In this method, the problem is the focus of the learning process and the guide in the inquiry process and in asking questions. In the section below, we comment on the development of the PBL process and the type of questions that students have asked themselves in this process.

### 4.1. PBL Process and Science Skills Development

The students completed the phases of the PBL process, in this case by deciding on their own problem to learn about—a real and ill-structured problem, contextualized in their environment [41]. They also design the inquiry process in order to establish a protocol to reach possible solutions to the problem. They were looking for solutions to the problem, evaluating them on a laboratory scale, developing experimentation protocols, taking data, repeating so results were reliable, etc. Finally, they presented the results of the inquiry to experts on the topic. Throughout the PBL process, they encountered difficulties in knowing where to turn in search of information, beyond what is usual for them (social networks). So, they needed a little guide to learn about programs like "We Are Scientists".

To verify whether this project has favored working inquiry in the classroom, we consider both the stages of inquiry proposed by Pedaste et al. [35] and the operations included in each stage, as described by Mosquera-Bargiela et al. [16]. The students asked a question that could be researched by the class group and on the school premises. The research topic stemmed from questions they asked through their observation of the environment. This led them to make a hypothesis (manure waste could be used as fertilizer for plants) and to create a complete design for experiments in order to test it. The students, after gathering

the necessary materials, performed the proposed experimentation by exploring, collecting data and interpreting the results they had obtained. They searched for an explanation for the unexpected results (discussing among themselves and also looking outside) and consulted experts on the matter. After interpreting the information collected, they were able to redesign a new sequence of experiments and introduce a new variable (the use of commercial fertilizer) as a control variable. This new variable enabled them to check if the results obtained with the fertilizer that they had prepared using manure produced similar results to the commercial fertilizer, as was the case. Once they had made this connection, they compared which was the most effective by studying the root mass and reaching conclusions based on all the information collected during the project. Finally, the students presented the results and conclusions to the teaching staff of the school and to the university lecturers that the school usually works with during the planning of this type of project. The science skills the students developed include identifying problems; formulating researchable questions; formulating hypotheses and predictions; designing and carrying out experiments; observing; measuring and collecting data; interpreting results; and preparing and communicating conclusions.

### 4.2. Questions

Analyzing the type of questions helps us to assess students' depth of learning. Most of the questions that the students asked themselves were deep, according to the classification scheme proposed by Graesser and Person [37]. These questions establish a causal link or antecedent, for example: "Why have not all the bulbs grown?", or "Why have the best results been obtained with the 0% manure solution?". They also asked questions concerning the objective of the project with the same depth, which involved generalist and systemic thinking. That means the students generalized the knowledge acquired throughout the process and much broader questions were raised, including questions that suggest systemic thinking, where the problem is part of a complex system, such as: "Could all livestock waste be used as agricultural fertilizer?", or "Would there be enough to replace commercial fertilizer?". These types of questions, which to be solved need to have a specific objective and are therefore specific, are difficult to ask if it is not the students themselves who do it [38]. Therefore, it can be concluded that the PBL context favors the posing of deep questions that involve establishing relationships between different types of knowledge and favor metacognition.

The students also ask intermediate questions in which they propose quantifying or specifying aspects of the process, such as: "How can we measure which of the two is the best fertilizer?" or "How many repetitions should we do to obtain representative results in each test?" There are some verification or comparison questions, such as: "Can we do research into an environmental problem in our environment?", or "Can manure be used as a fertilizer?"

We should state that, in general, questions with less depth were asked at the beginning of the project and, as the work advanced, the students' questions became deeper.

The students seemed very motivated at the beginning of the project because they were able to choose the research topic, which was an environmental problem in their immediate environment. Therefore, the students understood their work not only as an academic procedure to work on science but also as the search for a solution to a real and imminent problem. This motivation remained throughout the academic year and became a real and explicit interest in the work they were conducting. The fact that the students were free to study their own interests [12] and that the problems were contextualized in their immediate environment helped in turning this motivation into interest [8–11]. As De Pro [13] stated, these types of project make it possible to study not only content but also procedures, and, most importantly, they promote working with students on the asking of questions, which is the starting point for science learning [14–16].

## 5. Implications

In the present work, we have developed a PBL methodology to work on inquiry in science classes. Currently, the secondary education curriculum in Spain recommends working on science through learning situations, in which students practice inquiry. However, teachers sometimes lack the tools to design or implement learning situations. The PBL methodology encourages science learning through inquiry, and the present study is a real example of this.

**Author Contributions:** Conceptualization, J.P.-M. and E.C.-Z.; methodology, J.P.-M. and E.C.-Z.; software, J.P.-M. and E.C.-Z.; validation, J.P.-M. and E.C.-Z.; formal analysis, J.P.-M. and E.C.-Z.; investigation, J.P.-M., E.C.-Z., E.S.-S. and E.C.-S.; resources, J.P.-M., E.C.-Z., E.S.-S. and E.C.-S.; data curation, J.P.-M. and E.C.-Z.; writing—original draft preparation, J.P.-M. and E.C.-Z.; writing—review and editing, J.P.-M. and E.C.-Z.; visualization, J.P.-M., E.C.-Z., E.S.-S. and E.C.-S.; supervision, J.P.-M., E.C.-Z., E.S.-S. and E.C.-S.; project administration, J.P.-M. and E.C.-Z.; funding acquisition, J.P.-M. and E.C.-Z. All authors have read and agreed to the published version of the manuscript.

**Funding:** This research received no external funding.

**Institutional Review Board Statement:** The authors confirm all the subjects' participants in the present research have provided appropriate informed consent in a verbal form. We declare that the institution (and the country) in which we developed the research does not have an ethics committee; so, to develop the research, the authors requested the explicit consent of the students and their legal tutors.

**Informed Consent Statement:** Informed consent was obtained from all subjects involved in the study.

**Data Availability Statement:** Not applicable.

**Acknowledgments:** The authors would like to thank to the Beagle research group, to the IUCA research institute and to the PID2021-123615OA-I00 ministerial project for the support to this work.

**Conflicts of Interest:** The authors declare no conflict of interest.

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
