# Peer review of "Science Skills Development through Problem-Based Learning in Secondary Education"

_education, doi:10.3390/educsci13111096_

Round 1

Reviewer 1 Report

Comments and Suggestions for Authors

The manuscript clearly outlines the research problem. It focuses on the application of a Problem-Based Learning (PBL) methodology to foster inquiry in science classes and develop scientific skills. The context is set within the backdrop of an environmental issue, which is the use of manure as a potential fertilizer. The manuscript is highly relevant to the field of science education. It provides insights into the application of PBL in real-world classroom settings and its impact on student learning. The focus on environmental issues adds another layer of relevance, given the increasing emphasis on sustainability and environmental education. Overall, the manuscript is of quality and offers valuable insights into the field of science education. In light of this, here are some notes for improvement:

1.       While the manuscript does touch upon the benefits of PBL, it would be beneficial to provide a more comprehensive review of previous studies on PBL in science education. This will set the stage for the unique contributions of this study.

2.       Clearly articulate the gap in the existing literature that this study aims to address. This will help readers understand the significance of the research.

3.       Check your in-text citations. The author names are missing before the reference number. For example, page 1, line 32: “according to [7], this is not yet widely done. In view of all this, it seems clear that science 32”

4.       Delve deeper into any challenges or obstacles faced during the PBL process and how they were addressed. This will provide a more holistic view of the PBL experience.

5.       Provide a conclusion section for the article. Discuss the practical implications of the study for educators, policymakers, and curriculum designers. Offer recommendations for how PBL can be effectively integrated into science curricula.

6.       Clearly articulate the questions raised by students for future research, as mentioned in the manuscript. This will set the stage for subsequent studies and highlight the iterative nature of the PBL process.

Author Response

Dear editor:

We deeply appreciate the reviews that the reviewers have given us, which have undoubtedly helped to improve the quality of the article.

Below we answer to each of the comments and proposals of the reviewers and we also send the modified text with these proposals.

REVIEWER 1:

The manuscript clearly outlines the research problem. It focuses on the application of a Problem-Based Learning (PBL) methodology to foster inquiry in science classes and develop scientific skills. The context is set within the backdrop of an environmental issue, which is the use of manure as a potential fertilizer. The manuscript is highly relevant to the field of science education. It provides insights into the application of PBL in real-world classroom settings and its impact on student learning. The focus on environmental issues adds another layer of relevance, given the increasing emphasis on sustainability and environmental education. Overall, the manuscript is of quality and offers valuable insights into the field of science education. In light of this, here are some notes for improvement:

Thank you very much for your comments.

  1. While the manuscript does touch upon the benefits of PBL, it would be beneficial to provide a more comprehensive review of previous studies on PBL in science education. This will set the stage for the unique contributions of this study.

As recommended by the reviewer, we have expanded the theoretical framework in relation to PBL studies in science education contexts, adding a paragraph in the introduction, with new references. Therefore, we have modified the number of references in line with this change.

  1. Clearly articulate the gap in the existing literature that this study aims to address. This will help readers understand the significance of the research.

We thank the reviewer for this proposal, which undoubtedly focuses the objective of the study based on the importance of our work, thus we have added a paragraph to articulate the gap in the existing literature.

  1. Check your in-text citations. The author names are missing before the reference number. For example, page 1, line 32: “according to [7], this is not yet widely done. In view of all this, it seems clear that science 32”

We agree with the reviewer. Therefore, we have added the names of the authors when we refer to them directly.

  1. Delve deeper into any challenges or obstacles faced during the PBL process and how they were addressed. This will provide a more holistic view of the PBL experience.

As recommended by the reviewer, we have added a paragraph in the discussion section, in which we present the most relevant obstacles encountered during the PBL process.

  1. Provide a conclusion section for the article. Discuss the practical implications of the study for educators, policymakers, and curriculum designers. Offer recommendations for how PBL can be effectively integrated into science curricula.

As recommended by the reviewer, we have added an implications section where we discuss the practical implications of the study for educators and curricula.

  1. Clearly articulate the questions raised by students for future research, as mentioned in the manuscript. This will set the stage for subsequent studies and highlight the iterative nature of the PBL process.

As the reviewer indicates, we have added to the text the questions raised by the students for possible future research.

Reviewer 2 Report

Comments and Suggestions for Authors

Reviewer’s suggestions and comments on the Manuscript entitled:

Science skills development through problem-based learning in secondary education

Manuscript ID: education-2629752

This an excellent study regarding problem-based learning as a pedagogical approach to increase the popularity of science. The design of experiments, meteorology, results, and conclusions is excellent, instructive, and inspiring. An experiment covers all aspects of science: biology, chemistry, and even physics.

This manuscript has high citation potency. Therefore I recommend to the Editorial Office to accept this manuscript after minor revision.

Reviewer’s suggestions:

-          Lines 177/186: It is unclear how the solution of 100 % was made. What was the ratio between manure and water?

-          Figure 1: It is important that students are informed or self-informed about safety regulations in the lab. It can be noticed that the student is not wearing lab gloves and a white coat. So, good lab practice failed. Lab personnel had to warn students. Regarding education, this picture gives a bad example of professional behavior in the lab. Authors should take another picture showing proper lab dressing or exclude this picture. Everyone, especially young students has to be protected in the lab.

-          Table 1 gives information about the influence of manure % on the length and Figure 2 gives results about the influence of rooting agent, I don’t see a connection between Table 1 and Fig. 2

-          Lines 206/207 “We believe that the roots did not grow because they did not touch the water.” This is an honest admission of a huge mistake. Maybe it can be rewritten with the statement “because of poor contact with a solution”.

Author Response

Dear editor:

We deeply appreciate the reviews that the reviewers have given us, which have undoubtedly helped to improve the quality of the article.

Below we answer to each of the comments and proposals of the reviewers and we also send the modified text with these proposals.

REVIEWER 2:

Reviewer’s suggestions and comments on the Manuscript entitled:

Science skills development through problem-based learning in secondary education

Manuscript ID: education-2629752

This an excellent study regarding problem-based learning as a pedagogical approach to increase the popularity of science. The design of experiments, meteorology, results, and conclusions is excellent, instructive, and inspiring. An experiment covers all aspects of science: biology, chemistry, and even physics. This manuscript has high citation potency. Therefore I recommend to the Editorial Office to accept this manuscript after minor revision.

Thank you very much for your comments.

Reviewer’s suggestions:

-          Lines 177/186: It is unclear how the solution of 100 % was made. What was the ratio between manure and water?

We agree with this comment; it was not entirely clear what the 100% dissolution corresponds to. We have added the following sentence to the text to clarify: "This filtered liquid was considered 100% fertilizer concentration. And from that 100% sample, solutions in water were prepared."

-          Figure 1: It is important that students are informed or self-informed about safety regulations in the lab. It can be noticed that the student is not wearing lab gloves and a white coat. So, good lab practice failed. Lab personnel had to warn students. Regarding education, this picture gives a bad example of professional behavior in the lab. Authors should take another picture showing proper lab dressing or exclude this picture. Everyone, especially young students has to be protected in the lab.

We completely agree with the reviewer, and since it is not possible to take another photo because the project ended, we have deleted this image, and as a consequence we have modified the number of the figures. Thanks.

-          Table 1 gives information about the influence of manure % on the length and Figure 2 gives results about the influence of rooting agent, I don’t see a connection between Table 1 and Fig. 2

 The reviewer is right. We have had a confusion, therefore, we removed figure 2 from the text so that it does not lead to confusion. Thanks for identifying it.

-          Lines 206/207 “We believe that the roots did not grow because they did not touch the water.” This is an honest admission of a huge mistake. Maybe it can be rewritten with the statement “because of poor contact with a solution”.

We appreciate very much this proposal. We have modified the phrase and it is now: “We believe that the roots did not grow because of poor contact with a solution.”

Round 2

Reviewer 1 Report

Comments and Suggestions for Authors

I appreciate you attending to my previous comments and doing the revisions needed. The paper is definitely looking better. (Quick note: please correct Hmelo-Silver's name on page 2, line 55. It's currently written as Hmerro-Silver).

Reviewer 2 Report

Comments and Suggestions for Authors

The authors have successfully answered all reviewer's suggestions. Good work!